

# Magnetic Airborne Survey - Geophysical Flight

E. Camara[1], S.N.P. Guimaraes[2]

[1] Brucelandair, Ontario, Canada.
[2] Prospectors Aerolevantamentos e Sistemas Ltda., Rio de Janeiro, Brazil

*Correspondence to*: Erick Camara (erickcamara@msn.com)

**Abstract.** This paper provides a technical review process in aerial acquisition of geophysical data the methods used in recent times, with emphasis for magnetometry. Generally speaking addresses the calibration processes of geophysical equipment and also the aircraft to minimize possible errors in measurements. The corrections used in data processing and filtering processes are demonstrated with some results as well as the evolution of these techniques in Brazil and worldwide.

**1. Introduction**

Geophysics is a Geoscience that involves the study of the Earth via physical measurements. In this context, there are many types of physical measurements that can be studied. Airborne geophysics involved one of these types of measurements. It uses airborne data to characterize larger areas with mineral exploration potential. Measurements are typically taken at a preliminary point of the exploration process, after the soil of the area has been classified.

The first geophysical method to utilize airborne research was the magnetic method. Discovered by Faraday, sec. XIX, the method was initially used by the USSR (current day Russia) in 1936 (Hood, 1969) and better adapted by America in 1940 (Hood, 1969). Both countries had a vested military interest in the technology, particularly for submarine applications. After some adaptations, another early flight was made in the US in 1944 using the Beech Staggerwing NC18575 (Morrison, 2004). The first geophysical airborne survey in Brazil occurred 60 years ago (1953) in the city of Sao Joao Del Rey, Minas Gerais

(Hildebrand 2004). It was conducted by the Prospec Company, which later became Geomag. The survey utilized both magnetic and radiometric methods. The fixed wing aircraft used in the survey was the PBY-5 (Catalina). It was equipped with a Fluxgate magnetometer, which measured the total magnetic field, in the tail of the aircraft (Hildebrand 2004). The system was totally analogic and constructed using electromechanical units and an infinite series of valves. All the data processing was done manually because, at that time, analogic data was recorded, tabulated, corrected, interpolated and

plotted on a cartographic base. The data were then presented in the form of a profile overlay on contour maps. All tracing was also manually completed.

**Figure 1 – Example of the C208B model geophysical acquisition aircraft. (Source: Camara, E. Author Private Collection, Sept 2014).**



The acquisition methods more commonly used in airborne processes are the magnetometric and gamma spectrometric methods. Both methods require a data acquisition height of approximately 100 m, which allows the survey to show the study area in high detail. The acquired data are processed to obtain images or maps of a region, where key areas are those with contrasting magnetic fields (magnetometric case) and radioelement levels (gamma spectrometric case). The features depicted

in geological mapping studies can then be used to determine intrusions, faults and lineaments associated with subsurface geology. They can also provide indications of depth anomalies and possible mineralization areas. Therefore, this method has significant economic value, particularly for mineral exploration.

Aerogeophysics technology development has undergone several cycles over the past five decades. The most important advancement has been the use of digital technology. However, another massive technological step was made via the use of

navigational systems, satellite positioning and GPS (Global Positioning System). This technology became available when the United States government opened their satellite signal GPS to commercial users in the late 80's (Hildebrand 2004). Consequently, the development of automatic aeromagnetic compensators, color plotters and Windows software, such as Geosoft from Oasis Montaj, soon followed.

## 2. Geophysical Method Magnetometry

The magnetometric method measures small intensity variations in the earth's magnetic field (Reitz and Milford, 1966). Thus, it measures rocks that exhibit variable magnetism, which are distributed in the earth's crust above the Curie surface (Sordi 2007). These variations are present in different types of ferromagnetic rocks, including magnetite and basalt. These rocks exhibit magnetic variations in terrestrial magnetic fields, magnetically active regions and high terrains (Werner, 1953).

Because of these multiple magnetic influences, airborne data must be validated, and both external and internal influences

must be removed from the data sets. Data removal is conducted using diurnal variation calculus (diurnal monitoring) and the internal terrestrial magnetic field (based on the International Geomagnetic Reference Field (IGRF) mathematical model) (Ernesto 1979).

The IGRF model is approved by the International Association for Geomagnetism and Aeronomy (IAGA). It is a group of coefficients developed using spherical harmonics (Gaussian coefficients and Legendre polynomials), and is semi-normalized

to the $10^{th}$ degree. Every five years, this model undergoes a recalculation process until a definitive model is developed for the next 5 years. This definitive model is called the Definitive Geomagnetic Reference Field (DGRF). The eleventh degree of these equations about the geomagnetic field model can be related to the spatial dimension of the Earth's surface magnetic anomalies (Backus et al. 1996). Other books and papers are dedicated to just these topics and can be using to review and referred (Airo, 1999; Barton, 1988; Boyd, 1970; Elo, 1994; Hjelt, 1973; Parkinson, 1983; Puranen and Puranen, 1977;

Reford and Sumner, 1964).





### 3. Air Localization System or Navigation

In the early stages, air navigation for airborne surveys was performed using an aerial topographic map or aerial photographs and a video camera, which aided in future planning and management analyses.

Now, new and improved equipment is available for geophysics applications. Since the 1950s, large companies have had

access to microwave signal emitters. Multiple emitters were installed on aircrafts, eventually becoming the Inertial Navigation System (INS) for large aircrafts. Combined with a gyroscope, INSs calculate aircraft position.

However, the INS has been largely replaced by the GNSS satellite. GNSS satellites are small, highly precise, relatively cheap, widely available and use little energy, giving them a distinct advantage over other systems. (Bullock and Barritt, 1989; Featherstone, 1995; Hakli, 2004; Haugh, 1993).

**3.1. GNSS**

The GNSS is currently composed of 31 satellites, which operate in orbit. After 2016, some satellites will provide network measurements. In 2000, the American government disabled the Selective Availability (SA) filter, which controls the GPS, resulting in an improved system precision.

**Figure 2 – Natural localization model for satellites in the GNSS system. (Source: "United States Government" Public domain, Official U.S. Government information about the Global Positioning System (GPS) and related topics 2014. http://www.gps.gov/multimedia/images/ - accessed October, 2014.)**

Antennas are arranged to capture two frequencies, but one is reserved for military use. However, by receiving both signals, the measurement does not suffer degradation caused by the ionosphere. After 2020, new satellites will send two civil signals

rather than only one.

**Figure 3 – Example of a localization system with Selective Availability (3a) and Non -Selective Availability (3b). (Source: "United States Government" Public domain, *Official U.S. Government information about the Global Positioning System (GPS) and related topics* 2014.http://www.gps.gov/systems/gps/modernization/sa/data/ - accessed October, 2014).**

**4. Equipment Used in Geophysical Airborne Surveys**

The equipment used in airborne data acquisition includes both on-board and off-board systems. Acquisition tools are extreme sensitive. However, new and improved technologies regularly become available.

On-board systems are known as Stinger Systems and are typically installed on the tail of the aircraft. The aircraft is then adapted to prevent any materials from influencing the measurements. For example, when conducting magnetic

measurements, the aircraft is assembled with the least possible number of metallic substances or surfaces. Sensors are typically installed in the aircraft's extremes, such as wing tips, so that mechanical or human factors do not affect the



measurements. Pilots must be wary of the performance loss caused by the addition of sensors to the wing tips as they affect aerodynamics.

Systems with off-board equipment typically carry the sensor, often called the bird, equipment below the plane. This requires precise flying and a high level of compensation to attain reliable data.

**Figure 4 – Model of a B2 Helibras aircraft with a sensor bird and VTEM antenna. (Source: Guimarães, S. Author Private Collection, june 2008.)**

### 4.1. Aeromagnetometer

Aeromagnetometers encompass two common types of airborne acquisition equipment. One measures the three components

10 of the magnetic field and is called a Fluxgate. The other measures the resulting component of the magnetic field, called the Total Field. Both provide increased precision of the resultant magnetic measurements.

**Figure 5 – Example of the Earth's magnetic field components, including the total magnetic field vector, which is measured by the equipment. (Source: Guimarães, S. Author Private Collection, march 2006)**

In Figure 5, $B_x$ is the north magnetic component of $\vec{B}$, $B_y$ is the east component of the magnetic field of $\vec{B}$ and $B_z$ is the depth component of the magnetic field of $\vec{B}$. In addition, $\overset{J}{i}$ is the inclination angle in the horizontal plane and $\overset{J}{d}$ is the angle between geographical north and the horizontal component of the magnetic field, called the magnetic declination. The combination of the north and east magnetic field components form a new component, deemed the horizontal, which is

20 represented by $B_h$ in Figure 5.

This apparatus is highly sensitive to magnetic field measurements. It is regularly used for mineral, oil and gas prospecting. Normally, it is mounted on the stinger, bird or wing tips. Its operation involves Cesium vapor, which is not radioactive. The most common type of magnetometer is the G822A from Geometrics Ltd.

### 4.2. GNSS Receptor

25 The GNSS receptor provides the geographical location of the aircraft based on a global satellite system. It works as a signal receptor. The real time corrections have a precision of ± 3 meters.

**Figure 6 – Model of satellite signal receptors adapted for geophysical survey aircrafts. (Source: Modified from Product Drawing. GPS Source http://www.gpssource.com/products/search/160, March 2015).**



### 4.3. Altimeter Radar

Altimeter radar is used to measure the height of the system above a terrain. It is used to maintain a constant height when collecting measurements. Over rugged terrain, the processor uses the filters to correct for data acquisition inconsistencies. The system is used to construct terrestrial digital models to compare with satellite image models, such as the Shuttle Radar

Topography Mission (SRTM).

**Figure 7 –FreeFlight TRA-3500 Altimeter Radar with a height limit of 2500 ft (approx. 750 m). (Source: (n.d.) Retrieved November, 4, From Http://www.seaerospace.com/terra/tri40.htm. Reprinted with permission as per email).**

### 4.4. Navigation Agnav/FASDAS

Navigation Agnav provides differential GPS corrections in real time, allowing for accurate knowledge of the aircraft position and simplified navigation.

### 4.5. Input and Data Storage

Data acquisition equipment works as a magnetic processor and compensator. A common unit is the DAARC 500 from RMS, which is both a data collector and recorder. It allows for a simpler operation and can use up to eight magnetometers with

15 three axes each. The magnetometers are linked to a 32-bit computer and use advanced mathematics to calculate aircraft interference, axis movements or other factors. Data are visualized in real time via a liquid crystal screen.

**Figure 8 – DAARC 500 in operation. (Source: Camara, E. Author Private Collection, Sept 2015.)**

### 4.6. Compensation System

The compensation system monitors aircraft movement and magnetic interference. It is commonly fitted on the Stinger. It instantaneously improves data due to compensation measurements. One sensor-based compensator system is the TFM 100-LN from Billingsley Magnetics, which uses a magnetic flux sensor.

**Figure 9 – RMS DAARC500 Compensation System. (Source: Modified from RMS 2015 Retrieved November, 2015 from http://www.rmsinst.com/images/DAARC500.jpg).**

### 4.7. Camera

Cameras are used to record and monitor the flight area. They also help with processing as they often allow system operators to verify interference after data collection.





### 5. Airborne Surveys: the Initial Calibration Process of Magnetometry

Survey technologies have specific degrees of precision, based on resolution and other parameters. Therefore, some devices require calibration and stabilization prior to surveying. Thus, each device used in a survey may require a specific calibration method.

Because the magnetometer is a piece of magnetic equipment, any ferromagnetic object in the aircraft, including the engine, can directly interfere with measurements (Hood 1969). However, the sensor layout of the aircraft should take this into consideration, as should the materials used to build the craft, which should be non-magnetic. Ferromagnetic materials in the aircraft structure should undergo a demagnetization process and then remain stagnant for a long period of time. This is because the airframe can become static and influence the data acquisition. The calibration and inference compensation of

magnetic equipment are typically conducted on a flight known as an FOM.

### 6. Technical Instructions for calibration flight and tests

### 6.1. Figure of merit (FOM)

A test flight is conducted to analyze the active magnetism compensation caused by the aircraft and its components, such as engine accessories, engine masses, avionics, current generated on the fuselage and other factors. It is tested in the project

area and can include the four selected headings N-S and W-E or different on headings based on the project. The test must include a parallel control and production lines, according to the project guidelines. The sum of the anomalies in the area is received by the magnetometer when the aircraft performs control movements in all three axes. These control movements includes a ± 20º Roll (Longitudinal), yaw ± 10 ° (Vertical) and ± 10 ° Pitch (Lateral). At altitudes of 3000 ft (914 m) or 4000 ft (1220 m), the incoming soil variations are typically low so that only the heading and maneuvers affect the test. The

variations are stored in the system and used for automatic compensation during future data acquisition projects. (Hood 1969)
If any change is made to aircraft equipment or any project parameters, a new FOM flight must be completed.

**Figure 10 - Model of the aircraft maneuvers performed during the FOM test. (Source: Modified from http://www.thevoredengineers.com/2012/05//-the-quadcopter-basics, free domain).**

**Figure 11 - Example of magnetic field measurement interference caused by aircraft maneuvers. (Source: Guimarães, S. Author Private Collection, May 2007).**

### 6.2. Clove-Leaf

The Clove-Leaf flight test shows the degree of change experienced in the system when the aircraft changes heading during a

data acquisition.





Generally, this variation should be zero. However, it can be stored in a coefficient and compensated for throughout the project.

The flight is conducted at specified height based on a planned heading and North-South East-West directions. After initial test flights, new headings can be determined and flown via the same coordinates.

**Figure 12 - Maneuver model performed by the aircraft in the clove-leaf test. (Source: Modified from https://www.ibiblio.org/hyperwar/USN/ref/ASW-Convoy/ASW-Convoy-2.html, free domain).**

### 6.3. LAG

This flight test is used for measuring the magnetic field variations in different acquisition directions using a magnetometer

sensor. This test also utilizes radar altimeter measurements.

Generally, an anomalous region (magnetic and density) is selected to verify data along two acquisition headings, such as a hangar, ship, steel bridge or a previously determined anomaly. The annotated acquisition time is taken into account when performing the mapping.

**Figure 13 - LAG test results model applied to magnetic measurements. (Source: Guimarães, S. Author Private Collection, May 2007).**

### 6.4. Altimeter Radar

The altimeter radar test is conducted at heights of 200 ft, 330 ft (100 m), 400 ft (121 m), 500 ft (150 m), 600 ft (182 m), 700 ft (213 m) and 800 ft (244 m). For benchmarking purposes, the 330 ft (100 m) test should be completed three times.

The altimeter radar is important for data acquisition because the elevation can directly interfere with concentrations count in certain situations. In addition, barometric equipment may change with pressure and temperature.

### 6.5. Drape

In mountainous regions, a drape (pre-determined flight height) or relatively flat terrain is recommended for 3D processing. This allows high mountain surface data to be more easily attained, superimposed and mapped at a higher quality. In this

case, the height flown to acquire the control lines set the production lines height. This method accounts for the aircraft performance in the flight environment. Each aircraft climbs and descends at different rates based on size and other factors (Bryant 1997).

**Figure 14 - (a) drape model applied to the acquisition and control lines (b) topography of the terrain (c) results of an acquisition**
**line flight with drape. (Source: (n.d.) http://www.terraquest.ca/wp-content/uploads/2014/05/surveycontours.jpg Retrieved October, 2014).**



### 6.6. Contour

Contour flights use the radar altimeter for data acquisition and are best suited for flat land or sea (Offshore), often by helicopters. The pilot uses the radar altimeter to maintain a constant height of 300 feet above the ground, reaching 500 feet if towing a bird (Hood 1969). On terrain with accentuated topographical variations, this process makes it difficult to maintain

the pre-determined altitude. Climbs and descents are based on the pilot's experience, which is largely based on the craft, equipment and terrain. Therefore, using multiple pilots for data acquisition will cause data inconsistencies and require manual correction.

### 7. Geophysical Measurement Corrections

### 7.1. Magnetic Field

Magnetometric method corrections are necessary in the acquired measurements to distinguish only the anomalous magnetic field of interest. In this case, that is the crust magnetic field. Therefore, observations are made during aerial acquisition of the total magnetic field (external and internal).

### 7.1.1. Diurnal Magnetic Monitoring - BaseMag

In general, diurnal magnetic monitoring (DMM) uses a ground magnetometer. This equipment is installed at a fixed position,

called BaseMag, located as far as possible from magnetic interference. It is typically installed at the airport, at a location outside the pre-determined interference, which aids in logistical measures and equipment security.

It has built-in GPS for synchronization with aerial data acquisition. DMM takes measurements of the total magnetic field, which includes the main magnetic field (inside the earth), external interference (magnetic variation of the sun due to interactions with solar winds) and the crustal magnetic field.

These are ad hoc measurements, and in a magnetic interference-free area, the crustal magnetic field is negligible. Therefore, the IGRF mathematical model can provide us with values related to the main magnetic field. Thus, monitoring must be conducted entirely outside the interference zone of the study area. Note that modern monitoring equipment has a range limit of 27 NM (50 km), which is decreased during magnetic storms (Reeves 2005).

**Figure 15 - Example of a day monitoring BaseMag station, which measures the magnetic field in parallel to an airborne geophysical acquisition site. (Source: Guimarães, S. Author Private Collection, Jan 2015).**

**Figure 16 - Example of the diurnal magnetic field curve acquired at a BaseMag station. (Source: Guimarães, S. Author Private Collection, May 2007).**



### 7.1.2. Magnetic Anomalies in the Surface

Magnetic anomalies are varied counts peaks. These peaks may be caused by railways, power lines, magnetic storms, large metallic masses, ships, buildings and hangars. In addition, anomalies can be caused by equipment aboard the aircraft that contains chemical substances, which may be detectable by the instrument. (Hood 1969).

5 These peaks are clearly observed in the data. However, they must not be confused with magnetic anomalies caused by the subsurface of interest. These peaks should be filtered and removed from the data sets.

### 7.1.3. Diurnal Variations

During the day, the earth is bombarded with charged magnetic particles via solar winds. These loads compress from day to night and then expand, causing regular variations in the magnetic field. Nights are calmer for data acquisition, but more 10 impractical in certain regions. These variations are monitored via Basemag.

### 7.1.4. Magnetic Storm

Protons, electrons and accelerated atomic particles are a result of solar activity and are carried by solar winds, particularly during magnetic storms. These events can last for minutes or hours and may reach 90 km/h during geomagnetic storms. In some cases, the atmosphere may take days to stabilize. They have a larger influence at the earth's magnetic poles, affecting 15 GNSS signal reception and radio electronic equipment. This causes major issues for data acquisition.

Weather monitoring equipment provides alerts for large storm events. Generally, monitoring data and forecasts from meteorological research centers shown are consulted prior to flights. The most common weather study centers are the National Oceanic & Atmospheric Administration (NOAA), National Aeronautics and Space Administration (NASA) and their interagency branches.

### 20 8. Considerations Related to Geophysical Flight

Flights require the extreme attention of the crew. In addition to flying the aircraft, the pilot must monitor instruments and navigate. The pilot must simultaneously note the relation of the aircraft to land, cities, airports, air traffic, animals and other factors.

Normal flights follow pre-determined standards, such as the acquisition speed needed to preserve data resolution. Exceeding 25 this lateral limit (cross track) can cause an overlap of the perpendicular line, thus creating a gap on the map.

When approaching an obstacle, such as the ground, or simply following the drape, the pilot must anticipate the aircraft stabilization factors that can affect the propeller and flight path. When the power lever is increased to accelerate, the flow of air causes the craft to rise and tend to the left. Conversely, a decrease in the power level will decrease speed and cause the craft to tend to the right. This relationship is known as the P-factor, which affects cross tracking. It is most noticeable in



single-engine aircraft. The pilot should be alert to sudden power lever changes, which could lead to oversteering or overcompensation.

### 8.1. Line interception

The pilot may be given certain control lines to be flown. He may then consider the distances and degrees that allow the lines

to be most efficiently flown. EG a line on the bow with an NS curve to the right. It begins to curve 1.8 km away, with a stable tilt of 20 to cross the bow at 090. Note that 900 m is the distance at which the number is lower due to the curve. If it is greater, the pilot can choose to maintain or decrease the ratio by a few degrees. When flying LO head to cross bow 0 (360), a distance of 900 m is typically considered.

**Figure 17 - Representation of the control and provisional acquisition lines. (Source: Urquhart, W. 2013 Retrieved October, 2014 from http://www.geoexplo.com/flight_plan.gif. Reprinted with permission as per email)**

### 8.2. Flight Lines

A study area is divided into a network of lines in the North-South direction, commonly known as tie lines (cross - control),

and East-West direction, known as control lines. These lines are based on pre-determined requirements. Control lines may be located every 250 m to 1000 m for precision, whereas control lines can be spaced anywhere from 5 km to 10 km.

### 8.3. Completing lines

Various lines or line-segments may be flown successfully or unsuccessfully. These failures can include control lines or cross lines between control lines, which are most notable lines due to their typically large flight distance.

### 9. Examples of Results

During airborne geophysical acquisitions, it is necessary to conduct data quality checks. In general, Quality Assessment and Quality Control (QA-QC) are conducted on magnetometry and gamma spectrometry data, which are limited by lateral offset and acquisition speed. Parameters that undergo QA-QC analyses include the magnetic field, temperature, spectrum range and others. In addition, the acquisition area and control area are generally broken into grid blocks. Figure 20 illustrates the

25 quality of two types of data acquisition. Figure 20(a) was measured during the 1980's, when measurement equipment was much less sophisticated. Figure 20(b) was measured in 2005, with 250 m line spacing and using the latest equipment. Both refer to the same area, located in the southern portion of Minas Gerais, Brazil.





**Figure 18 - (a) Geophysical Brazil Germany Project acquisition (code 1009 -. CPRM, 1980) and (b) area 2 acquisition (Source: Guimarães, S. Author Private Collection, Nov 2012).**

Although a complete database was unavailable for Figure 24(b), the observed level of detail is much higher than in Figure
5    24(a). Note that developments in the airborne geophysics field have led to exponentially improved data, in terms of both coverage and quality. These data have allowed for significant mineral exploration, geological studies and geophysical analyses in Brazil and across the globe. For example, Figure 25 illustrates a subsurface map of high resolution aeromagnetic data, where the degree of certainty decreases as the data resolution increases.

10   **Figure 19 – Subsurface magnetic Field behavior based on aeromagnetic data. Location of magnetic sources of interest (Guimarães, Ravat and Hamza 2014).**

Others studies show results of these evolution process of the airborne data geophysical acquisition, can cite a few examples in the scientific literature as: Ravat, 1996; LaBrecque et al., 1997; Brozena et al.,2002 and 2003; Finn and Morgan, 2002; Salem and Ravat, 2003; Hinze et al., 2005; Hemant et al., 2007; Bouligand et al., 2014; Guimaraes et al., 2014.


## 10. Final Considerations

Increased geological knowledge and the development of new technologies, especially within information technology, have brought about important advancements in the study of Earth Sciences. The use of sensors for measuring different physical properties of minerals and rocks in mining has led to significant data improvements. These advances have allowed

geophysical surveys to become an essential part of mineral exploration and other fields.

The evolution of geophysical equipment and measurement systems has caused significant improvements in air data acquisition and quality. Thus, creating improved interpretative maps with economic geology implications has aided mineral exploration worldwide. This is thanks to improved magnetic field maps, radiometric, gravity and electromagnetic data, remote sensing and other data collection and processing methods.

This initial work was aimed at creating a summary of acquisition activities, including equipment and technical operations used to enhance geophysical measurements and associated results, as well as minimize problems encountered with these types of measurements.

### About the authors:

Camara, E. is a Chief Pilot employed by geophysical companies who has conducted geophysical airborne surveys since

2011, including work in South America, Central Africa and Asia, and was the recipient of Master Ground Instruction certification in the United States and Master certification in the Federal Aviation Administration's WINGS program.

Guimaraes, S.N.P. holds a Geophysics Ph.D. from the National Observatory/University of Kentucky, has worked with geophysical airborne analyses since 2006, and has worked on numerous geophysical projects aimed at unifying and

improving potential field data collected via airborne acquisition using geothermal resources.





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



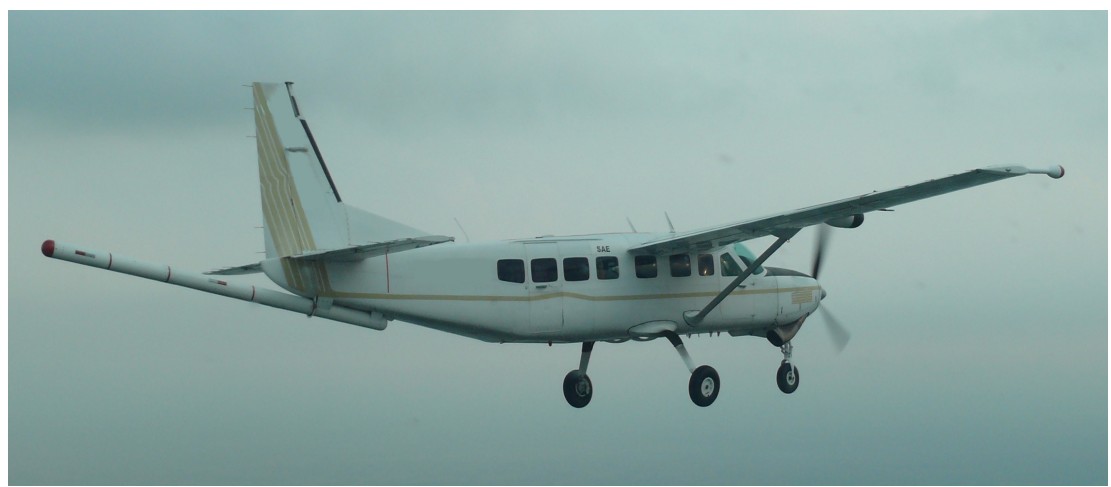

**Figure 1 – Example of the C208B model geophysical acquisition aircraft. (Source: Camara, E. Author Private Collection, Sept 2014).**



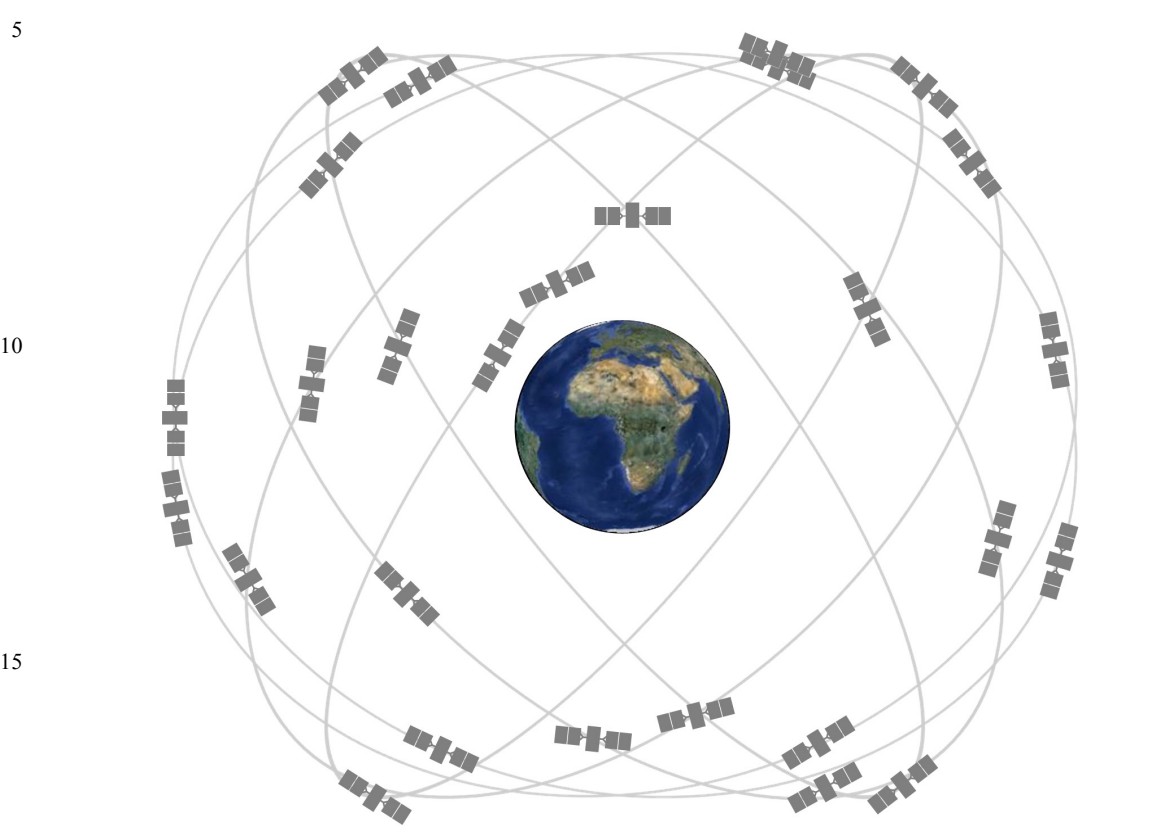

**Figure 2 –**

**Natural localization model for satellites in the GNSS system. (Source: "United States Government" Public domain, Official U.S. Government information about the Global Positioning System (GPS) and related topics 2014. http://www.gps.gov/multimedia/images/ - accessed October, 2014.)**



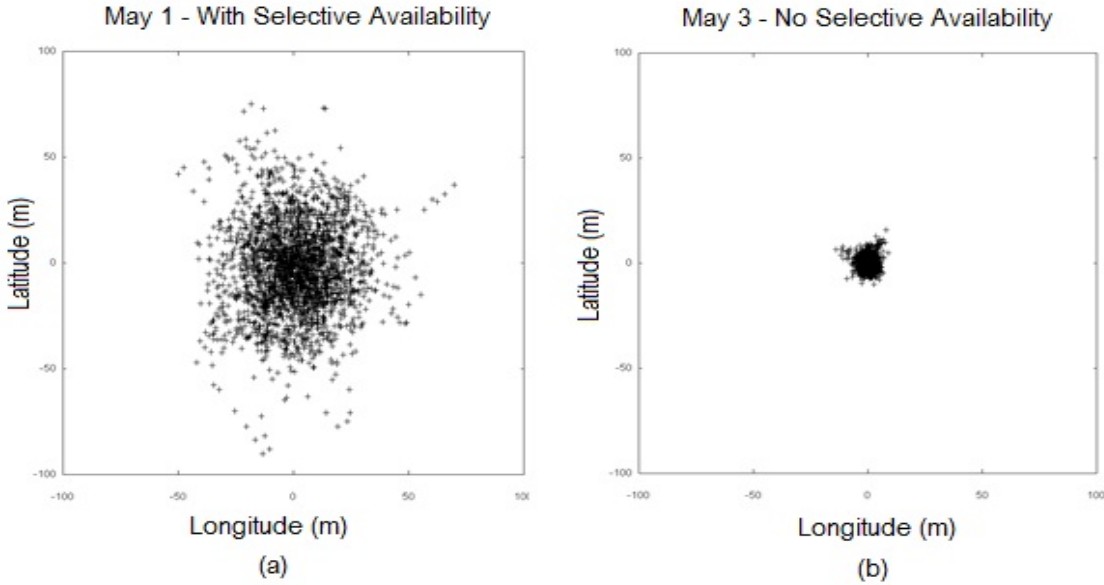

(a)

(b)

**Figure 3 – Example of a localization system with Selective Availability (3a) and Non -Selective Availability (3b). (Source: "United States Government" Public domain,** *Official U.S. Government information about the Global Positioning System (GPS) and related topics* **2014.http://www.gps.gov/systems/gps/modernization/sa/data/ - accessed October, 2014).**



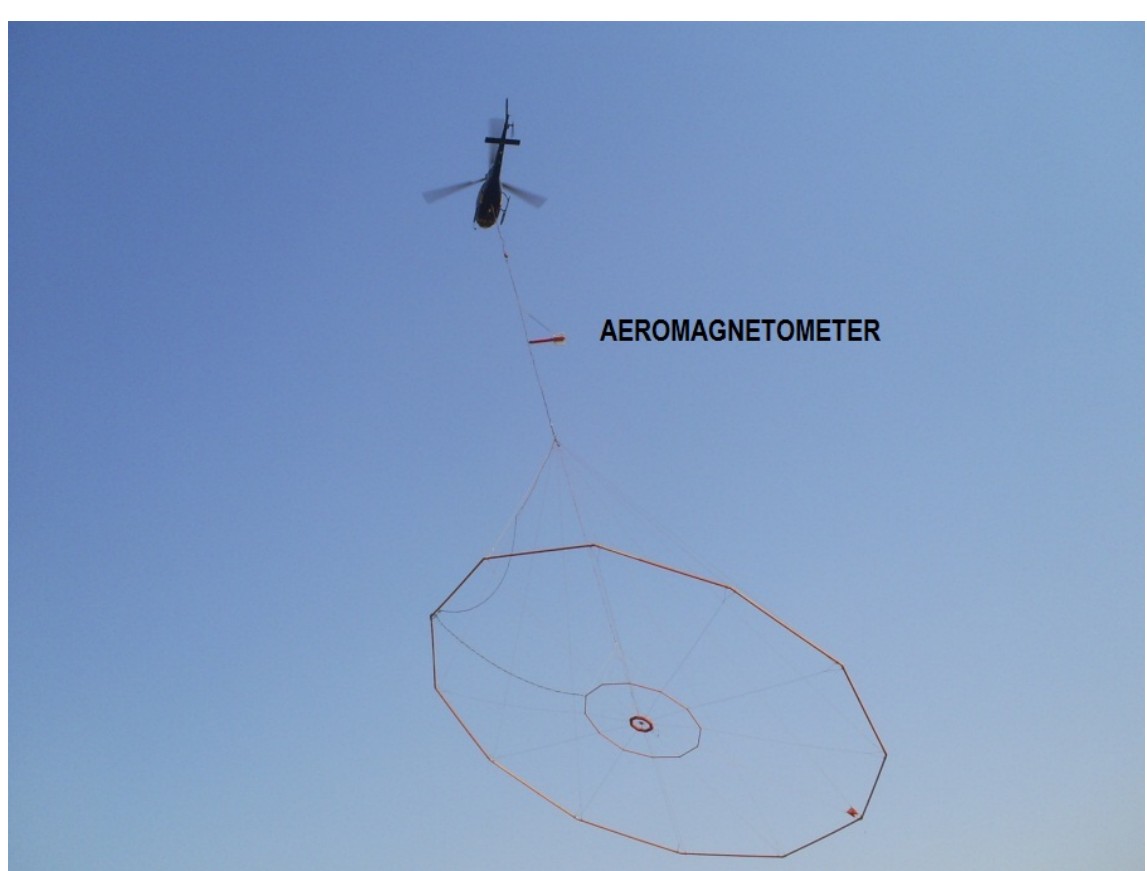

5   **Figure 4 – Model of a B2 Helibras aircraft with a sensor bird and VTEM antenna. (Source: Guimarães, S. Author Private
Collection, June 2008.)**





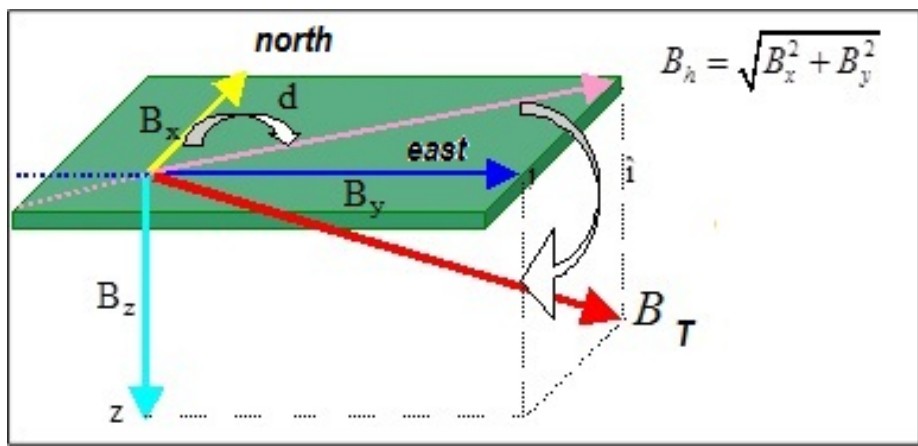

**Figure 5 – Example of the Earth's magnetic field components, including the total magnetic field vector, which is measured by the**
10   **equipment. (Source: Guimarães, S. Author Private Collection, march 2006)**



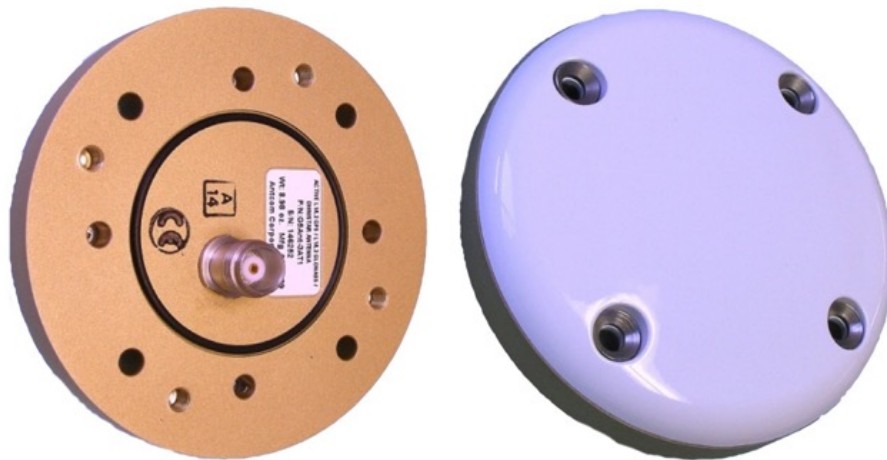

**Figure 6 – Model of satellite signal receptors adapted for geophysical survey aircrafts. (Source: Modified from Product Drawing. GPS Source http://www.gpssource.com/products/search/160, March 2015).**



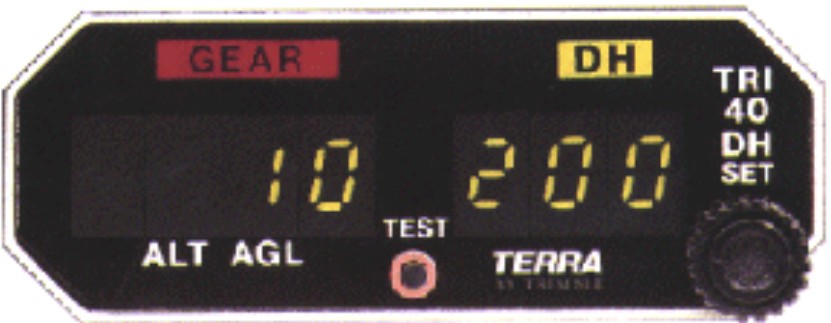

**Figure 7 – FreeFlight TRA-3500 Altimeter Radar with a height limit of 2500 ft (approx. 750 m). (Source: (n.d.) Retrieved
10   November, 4, From Http://www.seaerospace.com/terra/tri40.htm. Reprinted with permission as per email).**



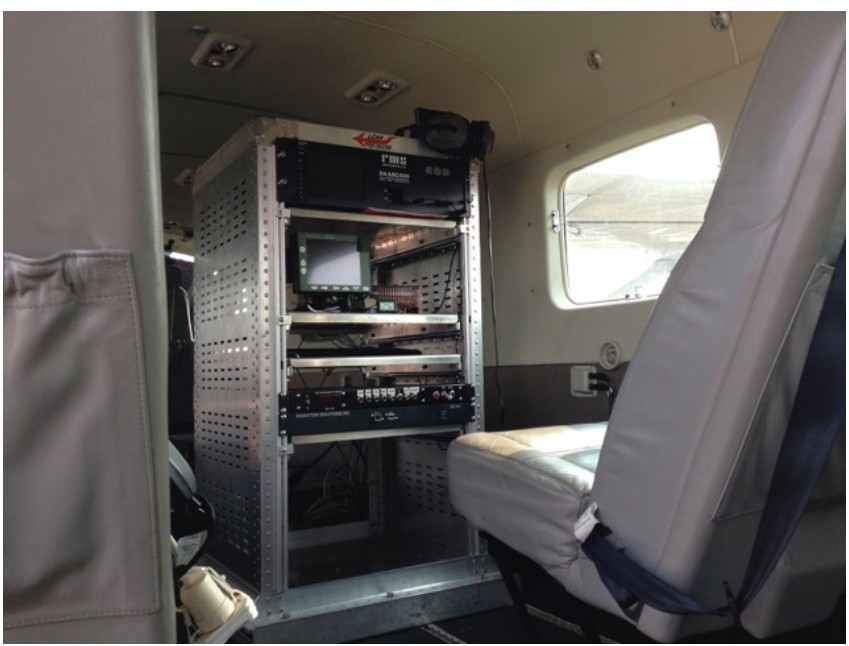

**Figure 8 – DAARC 500 in operation. (Source: Camara, E. Author Private Collection, Sept 2015.)**

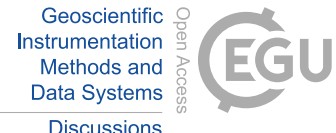

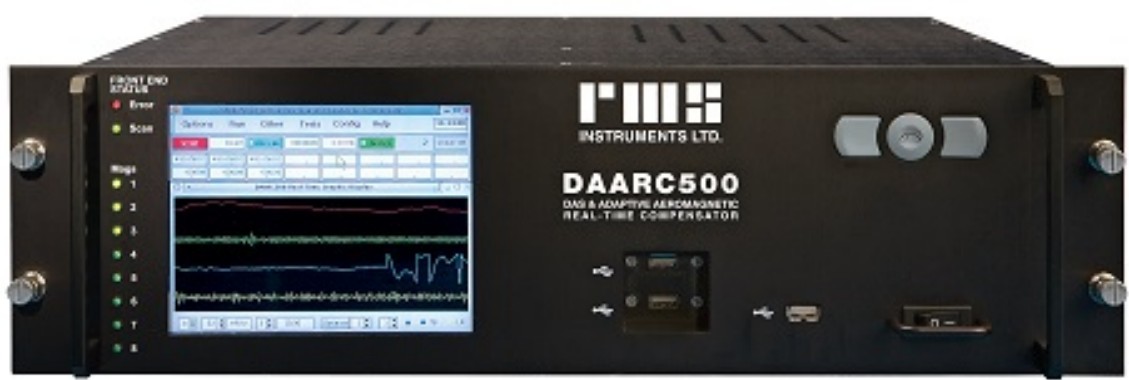

**Figure 9 – RMS DAARC500 Compensation System. (Source: Modified from RMS 2015 Retrieved November, 2015 from**
10 **http://www.rmsinst.com/images/DAARC500.jpg).**



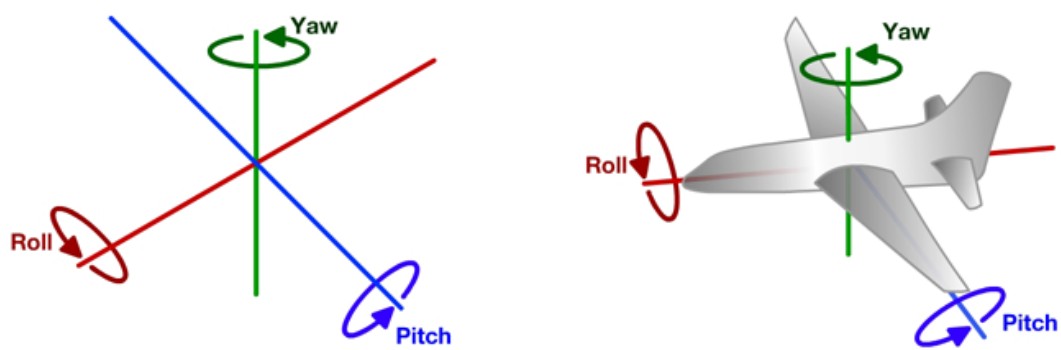

**Figure 10 - Model of the aircraft maneuvers performed during the FOM test. (Source: Modified from http://www.thevoredengineers.com/2012/05//the-quadcopter-basics, free domain).**







**Figure 11 - Example of magnetic field measurement interference caused by aircraft maneuvers. (Source: Guimarães, S. Author**
10   **Private Collection, May 2007).**





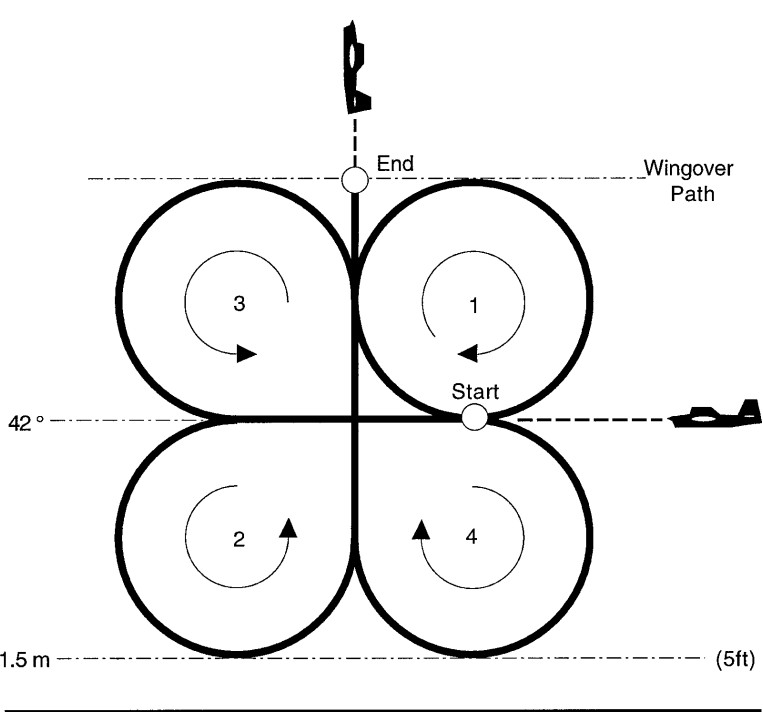

Four Leaf Clover

5  **Figure 12 - Maneuver model performed by the aircraft in the clove-leaf test. (Source: Modified from https://www.ibiblio.org/hyperwar/USN/ref/ASW-Convoy/ASW-Convoy-2.html, free domain).**



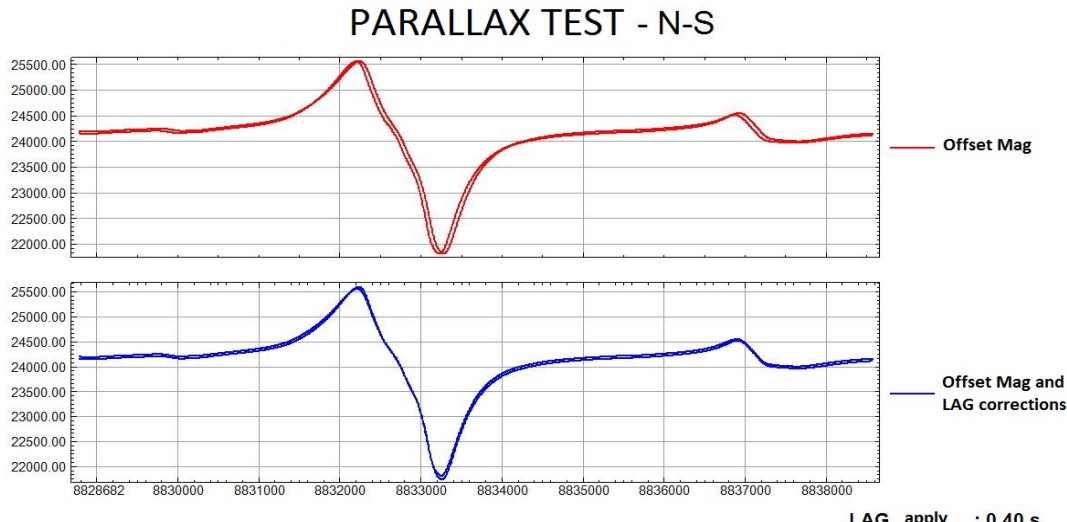

**Figure 13 - LAG test results model applied to magnetic measurements. (Source: Guimarães, S. Author Private Collection, May 2007).**




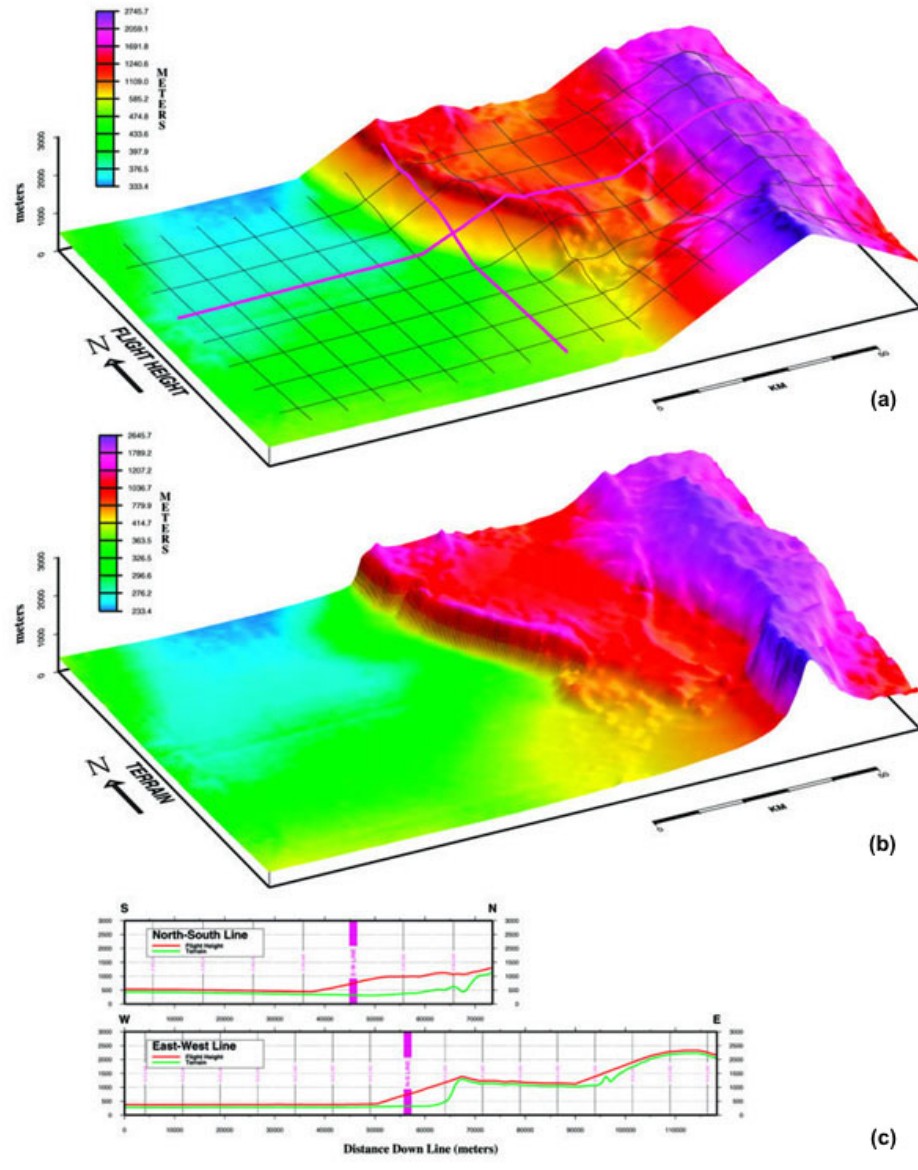

**Figure 14 - (a) drape model applied to the acquisition and control lines (b) topography of the terrain (c) results of an acquisition line flight with drape. (Source: (n.d.) http://www.terraquest.ca/wp-content/uploads/2014/05/surveycontours.jpg Retrieved October, 2014).**

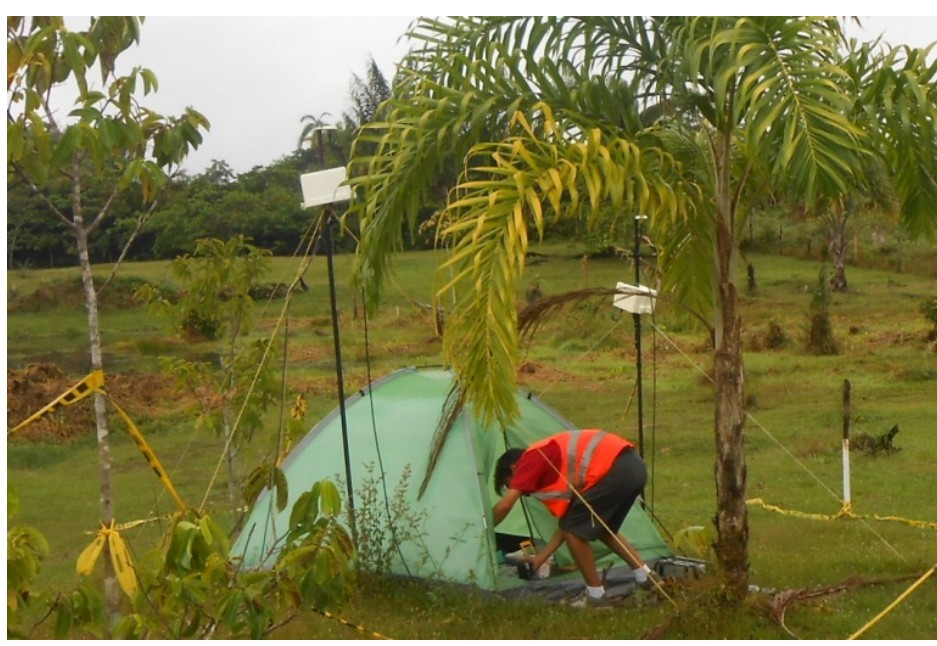

**Figure 15 - Example of a day monitoring BaseMag station, which measures the magnetic field in parallel to an airborne geophysical acquisition site. (Source: Guimarães, S. Author Private Collection, Jan 2015).**





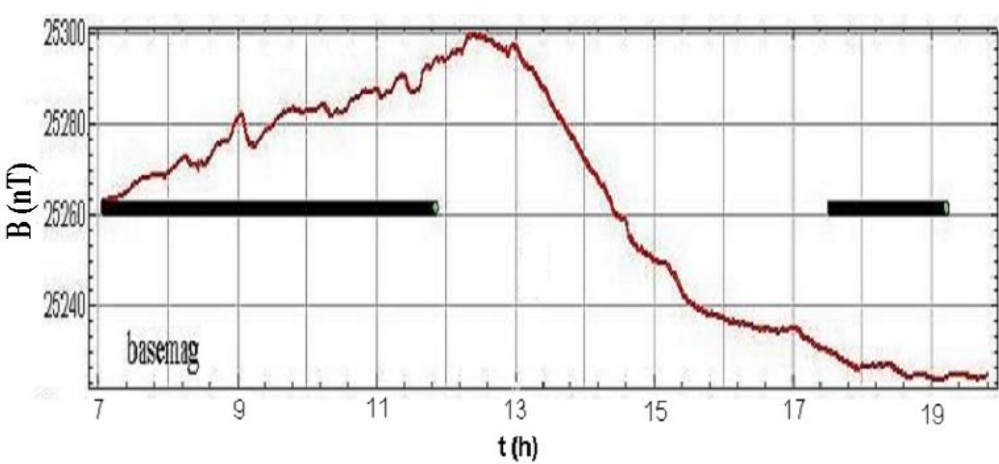

**Figure 16 - Example of the diurnal magnetic field curve acquired at a BaseMag station. (Source: Guimarães, S. Author Private**
10  **Collection, May 2007).**





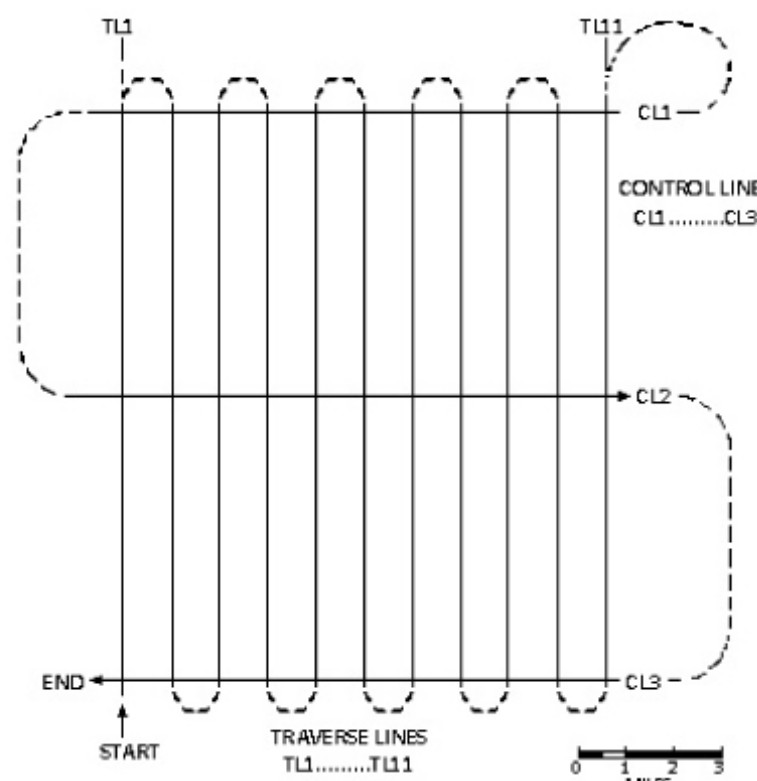

**Figure 17 - Representation of the control and provisional acquisition lines. (Source: Urquhart, W. 2013 Retrieved October, 2014 from http://www.geoexplo.com/flight_plan.gif. Reprinted with permission as per email)**





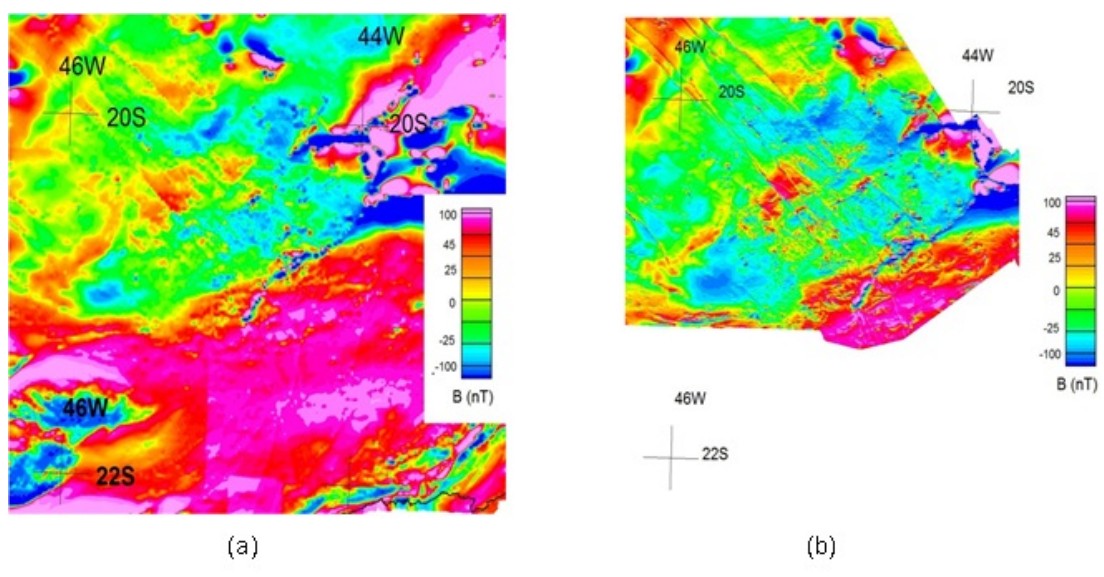

(a)                                        (b)

**Figure 18 - (a) Geophysical Brazil Germany Project acquisition (code 1009 -. CPRM, 1980) and (b) area 2 acquisition (Source: Guimarães, S. Author Private Collection, Nov 2012).**



**Figure 19 – Subsurface magnetic Field behavior based on aeromagnetic data. Location of magnetic sources of interest (Guimarães, Ravat and Hamza 2014).**