# Peer review of "Magnetic Airborne Survey - Geophysical Flight"

_Geoscientific Instrumentation, Methods and Data Systems, 2015_

## Referee Comment (RC1) · J. Lyrio (Referee) · 24 Mar 2016

The manuscript intends to provide the reader with a review of the instruments and procedures involved in an airborne magnetic survey. Overall the manuscript is poorly developed and contains several conceptual mistakes. The review of the fundamentals of the magnetic method are not entirely correct and the references provided do not take the reader to any textbook or classical paper in magnetometry. The majority of the paper is a disconnected sequence of poorly and careless descriptions of instruments and procedures used in an airborne magnetic survey. In my opinion, such text format makes hard to the reader not familiar with magnetic surveys to understand the complete operation. The final considerations neither discuss nor highlight the most important aspects in the manuscript. The references are incomplete and include items that are not mentioned in the text. Some of the figures are poor in quality and

legends mostly just describe what is in the figure without naming their main elements or even calling the attention to their principal aspects. I believe this manuscript is not ready for publication. It will be necessary a very strong review or, perhaps, a new start from scratch. Corrections, comments and suggestions were made in order to help the authors in improving the paper. Due to the large amount of these comments, suggestions and corrections a file was prepared using the Okular free-software to show the alterations on top of the original PDF file, without making real changes in the original file. Due to restrictions in uploading the file, I changed the original file extension from .okular to .zip. Sincerely, Julio Lyrio

Please also note the supplement to this comment:
http://www.geosci-instrum-method-data-syst-discuss.net/gi-2015-45/gi-2015-45-RC1-supplement.zip

---

## Referee Comment (RC2) · J. Lyrio (Referee) · 30 Mar 2016

[referee-annotated manuscript omitted]

---

## Author Comment (AC1) · 31 Mar 2016

Hello Professor, we are tankful for your revision, and we will consider some of it, including the figures numbers that are in excess. However, the opinion it has already been done, we will not send corrected version unless for possible consideration. On the other hand, we regret the analysis made by you since the paper has a technical nature, and is intentionally to beginning readers in the acquisition airborne geophysics process, since for this type of work to be done does not require an academic training in geophysics and even so, in a simpler language so that the paper will arouse the interest of workers in the area including airborne pilots like myself on the subject and maximize theirs potential, since we could not find such type of text in the literature. Items like GPS satellite is different than normal aviation due the dual band received thus the reason to be mentioned, forms of interception and line position is due an uncommon

manner in aviation, just survey and crop-dusting share similarities, the P-factor for the pilot to account for in maneuvers as well. Before this text come into discussion it went thought a harsh grammar correction with Wiley publisher. Now In this current magazine we went through three adaptations made by a respected editor and corresponding teacher's own magazine by Prof. Dr. Eppelbaum, and the manuscript was adapted for publication, he guided us with more citations, picture reduction and some correction, all other were approved by. Also sent informally to researchers from the area (airborne geophysics) that considered it very interesting when it proposes like renowned Dr. Alan Reid stated, "this is a really well done summary of what aeromag surveying is all about. Great teaching material too".

So if there is no reconsideration, and we can not carry out major adaptations at this work we thank you for your time and again we appreciate the comments.

---

## Referee Comment (RC3) · J. Lyrio (Referee) · 1 Apr 2016

Dear Authors, I believe the review is made by two independent reviewers and I don't know what happens if the other reviewer has different opinion. Anyway, as I wrote before, my opinion is that it would be better if you restart the manuscript from beginning and submitt it as a new manuscript. For sure, both the nature and the public a paper is trying to reach are relevant subjects and I agree that a paper that intends to reach beginning readers does not need to be neither complex nor deep, but it is imperative to the paper to be correct and clear, and your paper has failed to achieve such objectives. If the descriptions and explanations in your paper are not clear even for someone like me that is quite familiar with the subject, imagine for someone that has never heard about it. Finally I would like to say that I respect the opinions of all the other colleagues, including those mentioned by you, but in this case I have to disagree with

them. Sincerely, Julio Lyrio

---

## Referee Comment (RC4) · Anonymous Referee #2 · 19 Apr 2016

As stated by another reviewer, this paper have several issues that must be addressed before even consider to publish it. Besides its very limited English (that could be easily fixed) there is no clear focus on subject neither a proper target audience. The text lacks continuity and fail to keep a proper level of depth on its subjects. There is also some misconceptions and poorly explained fundamentals and pictures - e.g. the caption for figure 2 states: "Natural localization model for satellites in the GNSS system". This picture is indeed a plain and naive illustration about the concept of satellite constellation as used on GNSS for media use, providing no relevant information to intended review. There are other pictures of poor quality and questionable contents that should also be replaced or entirely removed. I would strongly suggest the authors to take this material back for a deeper analysis and remodeling before submitting for publish.

---

## Short Comment (SC1) · 20 Apr 2016

A good overview of the current state of in-field technology and procedures. As a former photogrammetrist, it was interesting to see the updates to the systems and the data processing that produce more accurate results.

The details about the pilot's responsibilities and flying techniques was particularly informative.

---

## Author Comment (AC3) · 9 May 2016

We appreciate your comment, the intention of the paper is to be simpler text and an abroad view of the subject is been seen as workers and a student in the area that does not have a deep knowledge of the subject and be in everyone reach. Therefore we disagree with the reviewer since it is not specified neither the scientific incoherence nor either the contents. It's just your point of view.

---

## Author Comment (AC4) · 9 May 2016

Thanks for your kind words, this work is for those whom work in the Airborne Survey area, Pilots, Operators and Geophysics students alike, and does not possess a overall read about it's systems and calibration results. And that's the reason we have reached an 875 downloads so far.